# An Improved MobileNetV3 Mushroom Quality Classification Model Using Images with Complex Backgrounds

**Fengwu Zhu, Yan Sun** **, Yuqing Zhang, Weijian Zhang and Ji Qi ***

College of Engineering and Technology, Jilin Agricultural University, Changchun 130118, China; zhufengwu@jlau.edu.cn (F.Z.); syan1128@163.com (Y.S.); zhangyuqing0726@163.com (Y.Z.); zwj05252021@163.com (W.Z.)
* Correspondence: jiq@jlau.edu.cn; Tel.: +86-18844098599

**Abstract:** Shiitake mushrooms are an important edible fungus, and their nutrient content is related to their quality. With the acceleration of urbanization, there has been a serious loss of population and shortage of labor in rural areas. The problem of harvesting agricultural products after maturity is becoming more and more prominent. In recent years, deep learning techniques have performed well in classification tasks using image data. These techniques can replace the manual labor needed to classify the quality of shiitake mushrooms quickly and accurately. Therefore, in this paper, a MobileNetV3_large deep convolutional network is improved, and a mushroom quality classification model using images with complex backgrounds is proposed. First, captured image data of shiitake mushrooms are divided into three categories based on the appearance characteristics related to shiitake quality. By constructing a hybrid data set, the model's focus on shiitake mushrooms in images with complex backgrounds is improved. And the constructed data set is expanded using data enhancement methods to improve the generalization ability of the model. The total number of images after expansion is 10,991. Among them, the number of primary mushroom images is 3758, the number of secondary mushroom images is 3678, and the number of tertiary mushroom images is 3555. Subsequently, the SE module in MobileNetV3_large network is improved and processed to enhance the model recognition accuracy while reducing the network size. Finally, PolyFocalLoss and migration learning strategies are introduced to train the model and accelerate model convergence. In this paper, the recognition performance of the improved MobileNetV3_large model is evaluated by using the confusion matrix evaluation tool. It is also compared with other deep convolutional network models such as VGG16, GoogLeNet, ResNet50, MobileNet, ShuffleNet, and EfficientNet using the same experimental conditions. The results show that the improved MobileNetV3_large network has a recognition accuracy of 99.91%, a model size of 11.9 M, and a recognition error rate of 0.09% by the above methods. Compared to the original model, the recognition accuracy of the improved model is increased by 18.81% and the size is reduced by 26.54%. The improved MobileNetV3_large network model in this paper has better comprehensive performance, and it can provide a reference for the development of quality recognition and classification technologies for shiitake mushrooms cultivated in greenhouse environments.

**Keywords:** MobileNetV3-large; SE; PolyLoss; mixed data set; mushroom; quality classification

## 1. Introduction

Shiitake mushrooms are the second leading edible fungus in the world and are loved by people because of their unique aroma, rich nutritional value, and delicious taste [1]. According to statistics, the production of shiitake mushrooms in China in 2021 reached 12.9572 million t, accounting for 31.34% of the total amount of cultivated edible mushrooms, and it was the main edible fungus produced in China in 2021 [2]. Nevertheless, there are still problems to overcome regarding the extensive production of these mushrooms. Since shiitake mushrooms have different nutrient content due to differing quality, they

need to be effectively classified. Traditional manual sorting [3] is time-consuming and laborious, and the benefits of this strategy are minimal. With the aging of our country's population and the serious loss of the rural population, relying on traditional manual sorting is difficult to sustain long term. In addition, shiitake mushrooms of differing quality have different sorting criteria in terms of shape and color, and it is difficult to screen out fresh shiitake mushrooms of excellent quality by relying on traditional machinery and equipment. Therefore, it is necessary to propose a classification method for effectively identifying the quality of shiitake mushrooms.

In recent years, with the rapid development of deep learning technology, Yonis Gulzar et al. conducted a classification study of sunflower diseases using a public data set containing a total of 1892 images of healthy and infected sunflower leaves and flowers. They investigated the classification of sunflower diseases using five deep learning techniques. It was shown that models such as MobileNetV3 and EfficientNetB3 provide high performance while requiring relatively few training cycles [4]. Yonis Gulzar et al. proposed a deep learning method based on Mobilenet_2 as an improved model utilizing transfer learning for training. In all, 40 different fruits were classified and recognized. The results showed that the proposed model achieved the highest accuracy in recognizing different types of fruits. The proposed model can be deployed in mobile applications for practical use [5]. Poonam Dhiman et al. systematically reviewed papers on prediction, detection, and classification of citrus fruit diseases using machine learning, deep learning, and statistical techniques. The best techniques that were found to be superior to others in their respective categories were as follows: SVM in ML methods, ANN in neural network networks, CNN in deep learning methods, and linear discriminant analysis (LDA) in statistical techniques [6]. Normaisharah Mamat et al. focused their research in the field of agriculture on the use of automated image annotation techniques for oil palm fruit maturity classification. They developed a model using deep learning combined with a transfer learning technique to train the model based on a data set. The results showed that the annotation technique successfully and accurately labeled a large number of images [7]. Liu et al. used the improved YOLOX deep learning method to detect the surface texture of shiitake mushrooms, and the model size was reduced by more than half [8]. Zhang et al. combined deep learning methods with spectral analysis to propose an end-to-end spectral qualitative analysis model based on convolutional neural networks that was used to effectively identify 20 grape varieties [9]. Wang et al. proposed an improved potato soil and stone detection method combined with YOLOv4, which realized the rapid detection and effective removal of soil and stones from debris-containing potatoes after harvest [10] Andre Dantes de Medeiros et al. combined X-ray images and deep learning network models to achieve 91%, 95%, and 82% recognition accuracy in terms of seed internal tissue integrity, germination rate, and vitality, respectively [11]. At the same time, the recognition model was gradually developed in the direction of lightweight. Zhang et al. developed a stud posture detection system to achieve rapid detection and high-precision positioning of studs [12]. Zou et al. proposed a deep imitation enhanced learning (DIRL) framework to effectively solve the problem of low efficiency of deep learning when exploring large continuous motion spaces in the field of autonomous driving [13].

Deep learning has been widely applied and developed in the field of visual recognition, but it is less applied in mushroom quality recognition and classification. Accordingly, this paper proposes a mushroom quality recognition classification model based on an improved MobileNetV3_large network [14] to address the above problems. Due to the complex environment of greenhouse mushroom cultivation, complex backgrounds in captured images cause serious interference in the recognition of mushroom features by the model. For this reason, this paper proposes a new method for constructing a data set, which effectively alleviates this problem. The recognition performance of the model directly affects the harvesting efficiency of shiitake mushrooms. In this paper, we improve the recognition performance of the model through simple changes while reducing the network size. In order to save the time cost and arithmetic power for training the model, this paper

utilizes an improved training strategy instead of the original model training method. The above method is expected to represent a lightweight, simple, and efficient classification model for mushroom quality recognition. The model can be deployed to mobile devices, such as cell phones, computers, artificial intelligence robots, etc., in practical scenarios. It can provide valuable technical support for the application and development of shiitake mushroom harvesting strategies in the field of precision smart agriculture.

The following points summarize the contributions of this paper:

- A new method for constructing data sets to enhance the model's focus on subjects in complex background images is proposed.
- A mushroom quality classification strategy based on the MobilenetV3_large network model is introduced to classify mushrooms of different quality levels.
- An improved MobilenetV3_large network model for mushroom quality classification is proposed based on different training strategies to improve the recognition accuracy of the model while reducing the time cost and arithmetic power spent on training the model, such as data enhancement techniques, migration learning techniques, and replacing the loss function with a better one.
- The recognition performance of the model after replacing the SE attention mechanism in the MobilenetV3_large network model is compared with CBAM, CA, scSE, and improved SE.
- The recognition performances for mushroom quality classification of eight other popular deep learning models are compared with that of the improved MobilenetV3_large network model.

## 2. Data Collection and Processing

### 2.1. Data Source and Division

The image data for this test were collected from Wulipu Village, Shancheng Town, Meihekou, Jilin, China, on 30 June 2023. The acquisition device was a Huawei Nova 3i smartphone (Produced by Huawei manufacturer in Shenzhen, China, the software version number is 2.0.0.263 (C01E261R1P3)), which generates images of $3456 \times 4608$ pixels, and the picture format was jpg. The shooting method was horizontal from the side, and a total of 275 original images of shiitake mushrooms were collected. Local shiitake mushroom quality grading standards and harvesting principles combined with the guidance suggestions from the planting experts at this location were employed. Finally, the image data for shiitake mushrooms were categorized into three classes, including Grade I, Grade II, and Grade III. Because of the obvious differences in the appearance characteristics of different grades of shiitake mushrooms, the grades of shiitake mushrooms were classified according to the degree of openness of the mushroom gills, the curvature of the outline of the cap, and whether defects occurred in the surface shape. See Table 1 for details.

**Table 1.** Shiitake Mushroom Quality Classification Standards.

| Indicator Level | First Class | Second Class | Third Class |
|---|---|---|---|
| The degree of opening of shiitake mushrooms | Half open | Fully open | - |
| Mushroom cap contour curvature | Full | Flat | - |
| Appearance defect | No | No | Yes |

In order to avoid model interference caused by multiple shiitake mushrooms in the image, the sample picture was appropriately cropped in this study, and only one shiitake mushroom body was retained in each image. Example images are shown in Figure 1.

In order to improve the recognition accuracy and generalization ability of the model and to enhance the robustness of the model, the background data set, a mixed data set, and an enhanced data set for removing shiitake mushrooms were also constructed in this experiment, and each data set was divided into a training set, verification set, and test set according to the 8:1:1 ratio. The number of images in each data set is shown in Table 2.

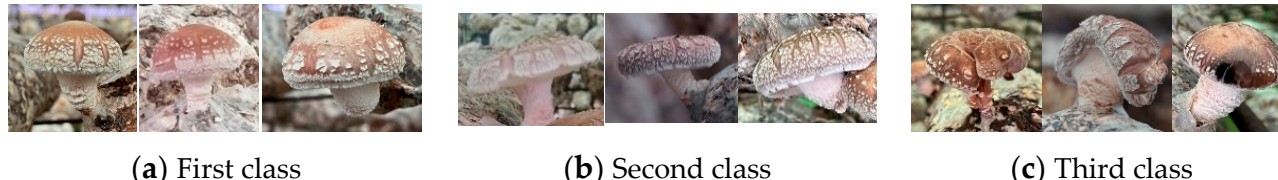

(**a**) First class　　　　　　(**b**) Second class　　　　　　(**c**) Third class

**Figure 1.** Original shiitake mushroom images.

**Table 2.** Statistical breakdown of the number of images in each data set.

| Data Division | Original Image/Removed Background Image | | | Mixed Data Set | | | Enhanced Data Set | | |
|---|---|---|---|---|---|---|---|---|---|
| | First Class | Second Class | Third Class | First Class | Second Class | Third Class | First Class | Second Class | Third Class |
| Training data set | 76 | 74 | 72 | 152 | 148 | 144 | 3007 | 2943 | 2844 |
| Verification data set | 9 | 9 | 9 | 18 | 18 | 18 | 376 | 368 | 355 |
| Test data set | 9 | 9 | 8 | 18 | 18 | 16 | 375 | 367 | 356 |

*2.2. Data Processing*

2.2.1. Background Segmentation

Since the background of shiitake mushroom images is complex, in order to reduce the interference of complex background information on model training and improve the recognition accuracy of the model, the background of the shiitake mushroom images was divided and removed in this experiment. Example mushroom images before and after removing the background are shown in Figure 2.

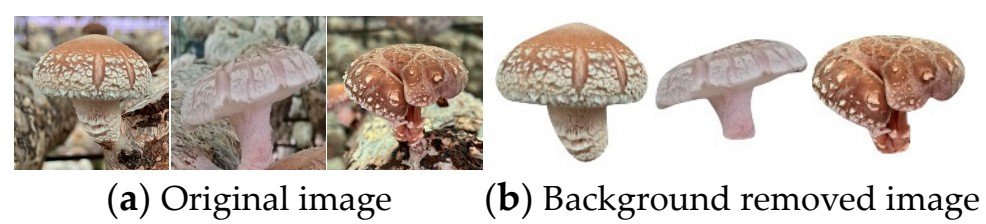

(**a**) Original image　　　(**b**) Background removed image

**Figure 2.** Mushroom images before and after removing the background.

2.2.2. Building a Hybrid Data Set

The shiitake mushrooms in the images with complex backgrounds and the images with the background removed represent the same type of shiitake mushroom when observed by the human eye, which stems from the subjective attention of humans on the shiitake mushrooms in the images. For computers, the shiitake mushrooms in the images with complex background and those with the background removed belong to two different types of images. Thus, complex background information may be recognized by the computer as a feature of the image, which in turn affects the training of the model.

In order to minimize the influence of complex background information on the model and increase the model's focus on mushroom ontology, this experiment constructs a hybrid data set to improve the accuracy of recognition by the model. In this paper, the data set composed of mushroom images with complex backgrounds and those with the background removed is defined as the hybrid data set. In this way, after the model recognizes two images with the same subject, it may pay more attention to the shiitake mushroom features in the image and reduce the interference of complex background information. Some of the images in the hybrid data set are shown in Figure 3.

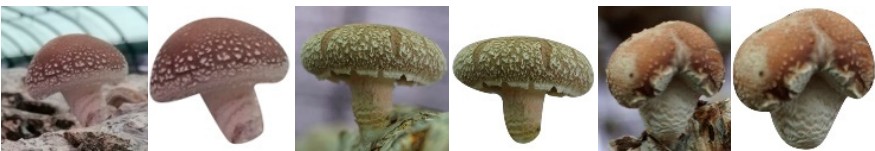

**Figure 3.** Examples of shiitake mushroom images in the mixed data set.

2.2.3. Data Enhancement

In order to further improve the recognition performance of the model, enhance the generalization ability of the model, and reduce the phenomenon of overfitting, in this experiment, a data enhancement strategy [15] was used to expand the mushroom image data in the mixed data set. Random rotation, random translation, random flipping, and random brightness transformation were used to expand the original data to 20 times the original, totaling 10,991 shiitake mushroom images. Among them, random rotation involved randomly rotating the image data in the range of 0~360°; random translation involved randomly translating the image data in the horizontal and vertical directions; random flipping involved randomly flipping the image data in the horizontal and vertical directions; and random brightness transformation simulated the color characteristics of shiitake mushrooms under different brightness levels by randomly adjusting the RGB value of the input picture between 50 and 100. Examples of the expanded shiitake mushroom images are shown in Figure 4.

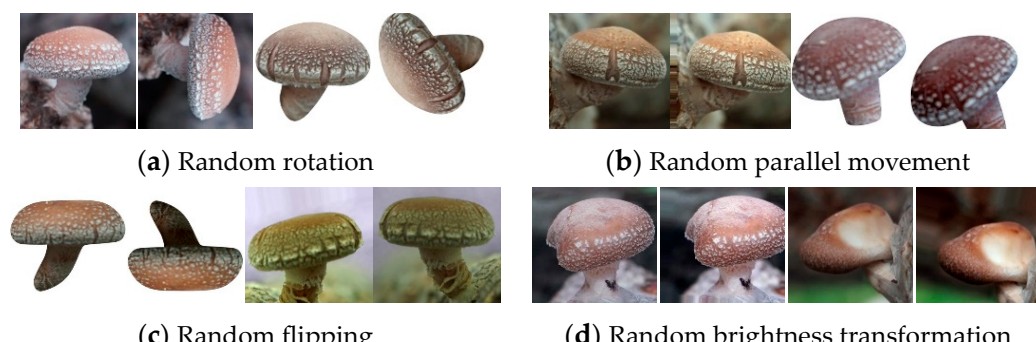

(**a**) Random rotation        (**b**) Random parallel movement

(**c**) Random flipping        (**d**) Random brightness transformation

**Figure 4.** Examples of shiitake mushroom images after data enhancement.

**3. Construction of a Quality Classification Model for Shiitake Mushrooms**

*3.1. MobileNetV3-Large Network Model*

The MobileNetV3 network improves on the MobileNetV2 network [16] by using a depth-separable convolutional structure and an inverted residual structure to ensure the model is lightweight; additionally, the SE module [17] and h-swish activation function are introduced, and the time-consuming layer structure in the bottleneck is redesigned, further improving the recognition accuracy of the model.

In this paper, the MobileNetV3-Large model is used, which consists of several components, including four 2D convolutional layers, two bottleneck layers (112 × 112), two bottleneck layers (56 × 56), three bottleneck layers (28 × 28), six bottleneck layers (14 × 14), two bottleneck layers (7 × 7), and one pooling layer (7 × 7). The activation functions used are Swish and ReLU. Also, the SE attention mechanism is embedded in the bneck structure in this network to improve the recognition performance and reduce the space occupied by the network. The SE mechanism assigns different weights to different channels of the feature map by compressing the excitation operation. Finally, a 2D convolutional layer (1 × 1) is used to obtain the feature vector to achieve the final classification output.

Table 3 shows the specific structure of the model. Input denotes the size of the input image. Operator denotes each step of the model's operation on the feature map. exp size denotes the number of output channels of the extended layer. #out denotes the number of output channels of the feature map after passing through each layer of the network.

SE denotes whether this attention mechanism is added to each layer of the network. NL denotes the activation function used by each layer of the network. s denotes the step size of the network operation in each layer.

**Table 3.** MobileNetV3-Large network structure.

| Input | Operator | Exp Size | #out | SE | NL | s |
|---|---|---|---|---|---|---|
| $224^2 \times 3$ | conv2d | - | 16 | - | HS | 2 |
| $112^2 \times 16$ | bneck, $3 \times 3$ | 16 | 16 | - | RE | 1 |
| $112^2 \times 16$ | bneck, $3 \times 3$ | 64 | 24 | - | RE | 2 |
| $56^2 \times 24$ | bneck, $3 \times 3$ | 72 | 24 | - | RE | 1 |
| $56^2 \times 24$ | bneck, $5 \times 5$ | 72 | 40 | $\checkmark$ | RE | 2 |
| $28^2 \times 40$ | bneck, $5 \times 5$ | 120 | 40 | $\checkmark$ | RE | 1 |
| $28^2 \times 40$ | bneck, $5 \times 5$ | 120 | 40 | $\checkmark$ | RE | 1 |
| $28^2 \times 40$ | bneck, $3 \times 3$ | 240 | 80 | - | HS | 2 |
| $14^2 \times 80$ | bneck, $3 \times 3$ | 200 | 80 | - | HS | 1 |
| $14^2 \times 80$ | bneck, $3 \times 3$ | 184 | 80 | - | HS | 1 |
| $14^2 \times 80$ | bneck, $3 \times 3$ | 184 | 80 | - | HS | 1 |
| $14^2 \times 80$ | bneck, $3 \times 3$ | 480 | 112 | $\checkmark$ | HS | 1 |
| $14^2 \times 112$ | bneck, $3 \times 3$ | 672 | 112 | $\checkmark$ | HS | 1 |
| $14^2 \times 112$ | bneck, $5 \times 5$ | 672 | 160 | $\checkmark$ | HS | 2 |
| $7^2 \times 160$ | bneck, $5 \times 5$ | 960 | 160 | $\checkmark$ | HS | 1 |
| $7^2 \times 160$ | bneck, $5 \times 5$ | 960 | 160 | $\checkmark$ | HS | 1 |
| $7^2 \times 160$ | conv2d, $1 \times 1$ | - | 960 | - | HS | 1 |
| $7^2 \times 960$ | pool, $7 \times 7$ | - | - | - | - | 1 |
| $1^2 \times 960$ | conv2d $1 \times 1$, NBN | - | 1280 | - | HS | 1 |
| $1^2 \times 1280$ | conv2d $1 \times 1$, NBN | - | k | - | - | 1 |

*3.2. Improving the MobileNetV3-Large Network Model*

The recognition model in this paper is mainly based on the MobileNetV3_large network architecture, in which some of the network layer structures in the SE module of the bneck structure are replaced and the parameters are tuned. The recognition performance of the improved MobileNetV3_large network model is slightly improved, the occupied space is slightly reduced, and the number of network layers remains unchanged after the improvement. Figure 5 shows the schematic diagram of the network structure after the model improvement.

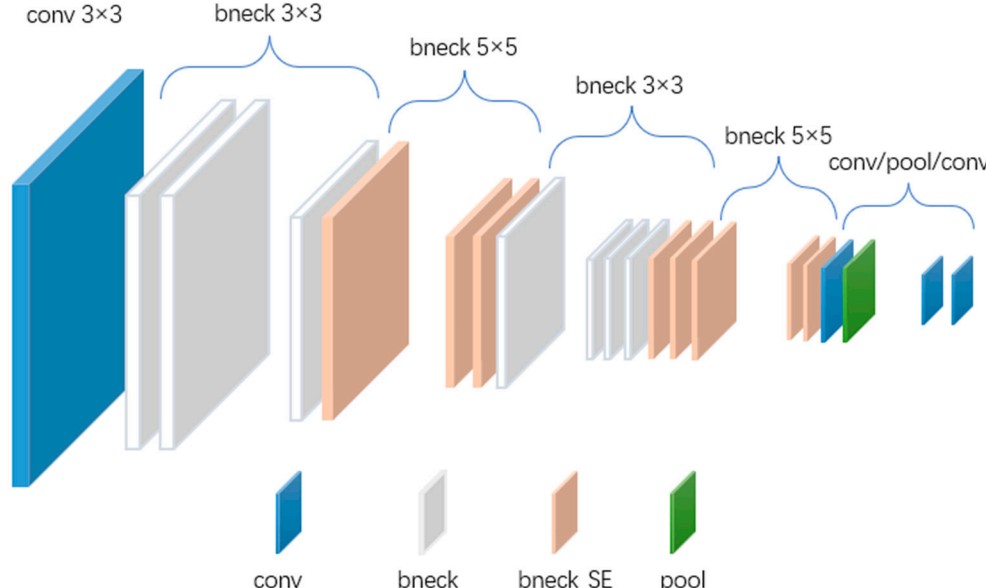

**Figure 5.** Schematic diagram of the improved MobileNetV3-large network structure.

### 3.2.1. Improved SE Module

In order to strengthen the relationship information between channels, in 2018, Hu et al. [17] first proposed the SE module, which is flexible in operation and simple in structure and can significantly improve the network performance by adding a small number of network parameters. The SE module is mainly composed of two parts, Squeeze and Excitation. First, the Squeeze operation is performed on the input feature map using global average pooling to compress the global spatial information into channel descriptors; then, the importance of each channel in the obtained aggregated information is predicted using the fully connected layer of Excitation to obtain the importance among different channels; finally, the channel weights of the output of Excitation are weighted onto the previous features channel by channel by multiplication channel-by-channel weighting to the previous features to complete the recalibration of the original features in the channel dimension.

In the MobileNetV3-Large network model, the authors embedded 8 SE modules in 15 bneck structures, so the influence of the SE mechanism on the network recognition performance and network size is significant, and optimizing the structure and parameters of the SE mechanism is especially important for improving the network performance. Therefore, this paper improves the SE modules in the bneck structure of the MobileNetV3-Large network model.

As the fully connected layer mainly compresses the features of the image, the convolutional layer mainly extracts the features from the image. After the image has gone through multiple convolutional layers, the features in the output feature map become more advanced. And after the feature map has gone through multiple fully connected layers, the features are compressed more centrally, and the weight that each feature has becomes more important. As the background of the mushroom image in this paper is more complex, the diverse backgrounds may also be recognized by the computer as multiple features of the image, which is obviously unfavorable for the recognition of the model. It is feasible to distinguish the grade of mushroom quality by shallow appearance features, and observing more advanced abstract features may be unnecessary. Therefore, the recognition model in this paper needs to filter out as many complex and diverse background features as possible so that the model can focus more on the more concentrated and important ontological features of shiitake mushrooms in the image. Then, replacing the convolutional layer with a fully connected layer may be a good choice. Therefore, in this paper, the convolutional layer in the SE module is replaced with a fully connected layer to enhance the output channel weights.

After replacing the convolutional layer with a fully connected layer, the network increases the number of computational parameters by a small amount in order to further improve the performance of the model and reduce the space occupied by the model in practical applications. In this paper, we minimize the size of the network by adjusting the compression multiplier of the channels in the SE module from $4\times$ to $16\times$ while ensuring model recognition accuracy. Compared with the original model, the improved model is smaller in size.

The SE module after the improvement allows for better recognition accuracy of the model, while making the model smaller. The structure of SE before and after improvement is shown in Figure 6.

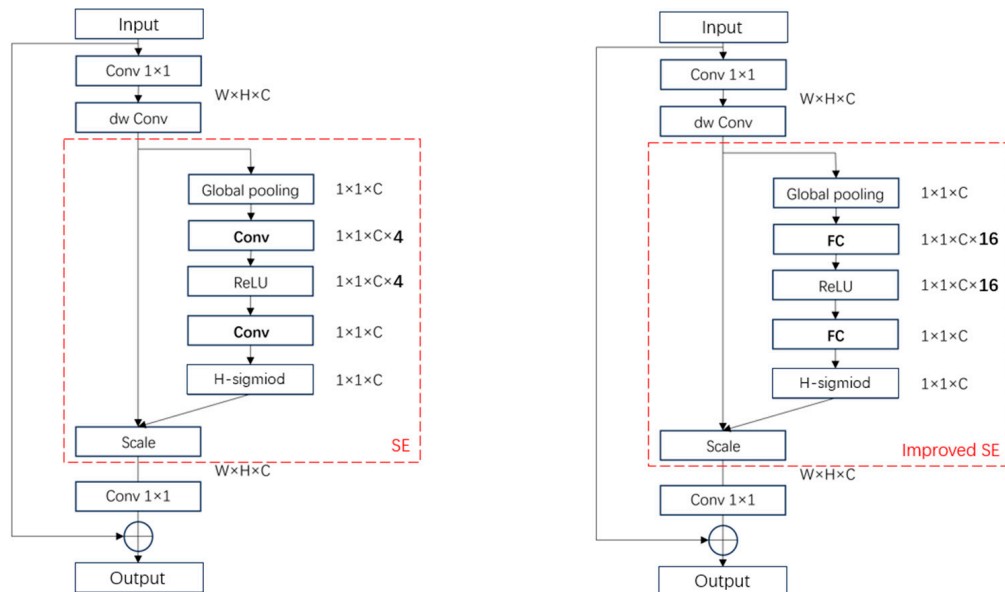

**Figure 6.** The original SE module (**left**) and the improved SE module (**right**).

### 3.2.2. PolyLoss

In order to be more flexible and applicable for the classification tasks of this experiment, the PolyLoss loss function is introduced. Inspired by Taylor's expansion to approximate the function, Leng Zhaoqi and others proposed PolyLoss, designing and treating the loss function as a linear combination of polynomial functions [18]. PolyLoss can adjust the weight of the polynomial loss through a simple strategy and a line of code to reduce the difficulty of adjusting the hyperparameters. At the same time, the cross-entropy loss function (CrossEntropyLoss) and focus loss function (FocalLoss) commonly used in the classification task of training deep neural networks can also be used as special cases of PolyLoss, and the application is more flexible.

The PolyLoss loss function decomposes the commonly used classification loss functions (such as cross-entropy loss and FocalLoss loss) into a series of weighted polynomials through Taylor expansion; the decomposition form is:

$$\sum_{j=1}^{\infty} \alpha_j (1 - P_t)^j \tag{1}$$

where (1) $\alpha_j \in R+$ represents the weight of the polynomial loss;

(2) $P_t$ represents the probability of the target label prediction.

When $\alpha_j = \frac{1}{j}$, PolyLoss is equivalent to cross-entropy loss. Simply adjusting the polynomial coefficient $\alpha_j$ according to the unused tasks and data sets can achieve better results than the cross-entropy loss function and the focus loss function. The specific formula is as follows:

$$L_{Poly} = \alpha_1 (1 - P_t) + \alpha_2 (1 - P_t)^2 + \cdots \alpha_N (1 - P_t)^N + \cdots = \sum_{j=1}^{\infty} \alpha_j (1 - P_t)^j \tag{2}$$

In most cases, adjusting the first polynomial coefficient has a significant gain on the model, and the formula is as follows:

$$L_{Poly-1} = (1 + \varepsilon_1)(1 - P_t) + \frac{1}{2}(1 - P_t)^2 + \cdots = -\log(P_t) + \varepsilon_1 (1 - P_t) \tag{3}$$

After testing, when $\varepsilon_1 = 1$, the model training effect is optimal, so this test sets the value of the superparameter $\varepsilon_1$ of the introduced PolyLoss loss function to 1.

### 3.3. Migration Learning

In order to enhance the generalization ability of the model, improve the recognition accuracy of the model, and reduce the time cost, data volume, and computing power required for training neural networks, this study uses a combination of a migration learning strategy and the improved MobileNetV3-Large network model to construct a mushroom quality classification network model. The migration learning strategy can effectively alleviate the overfitting problem caused by the small amount of sample data in the deep network model and can speed up the convergence of the model and improve the efficiency of the training model [19].

## 4. Test Results and Evaluation

### 4.1. Test Environment

The training of the network model in this experiment is based on the pytorch deep learning framework. The test hardware environment is as follows: the CPU uses Intel (R) Core (TM) i7-12700F clocked at 2.10 GHz, the memory is 32 GB, the GPU uses NVIDIA GeForce RTX4070, and the video memory capacity is 12 GB. Using the Windows10 operating system, the python version is 3.10.

In this experiment, the hyperparameters of the network are balanced and tuned. The total number of training rounds (epochs) is set to 60; the number of training batches per round (batch_size) is set to 64; and the learning rate affects the update speed of the network weights. When the learning rate is set to $1 \times 10^{-4}$, the training effect of the model is the best. By default, the Adam optimization algorithm [20] is used. The algorithm iteratively updates the network weights based on the training data, which is simple to implement, computationally efficient, and requires less memory.

### 4.2. Evaluation Indicators

In order to objectively evaluate the effectiveness of the test method, four evaluation indicators [21], namely, the recognition accuracy rate, accuracy rate, recall rate, and F1 score, were used to evaluate the improved MobileNetV3-Large network model proposed in this article.

### 4.3. The Impact of Data Processing on the Accuracy of the Model

Figure 7 shows the accuracy training process of the original MobileNetV3-large network model on different data sets. It can be seen that training on the original image data set has the worst effect on improving the recognition accuracy of the model. After removing the background data set training, the recognition performance of the model improves slightly. After training from the mixed data set, the accuracy rate of the model is significantly faster than that of the original data set and the background image data set. Using data enhancement strategies to expand the mixed data set, the training process of the model is smoother, and the recognition performance is significantly improved.

Table 4 shows the test results of training the model using different data sets. It can be seen that the model recognition effect trained from the data set constructed from the original picture is poor, and the test accuracy rate is 81.1%. Using a data set that removes the background of the image to train the model, the recognition accuracy of the model is improved, and the test accuracy rate is 86.8%. Using the mixed data set to train the model, the recognition accuracy rate of the model is further improved to 90.7%. Using the data enhancement strategy to expand the mixed data set, the recognition accuracy rate in the test set reaches 97.2%, and the model recognition effect obtained after training is the best.

The number of pictures in the data set constructed from the original picture and the image with the background removed is small, resulting in fewer pictures for testing after the model is trained. In order to objectively compare the effects of different data sets on the training model, this paper uniformly uses the test set divided by the mixed data set to perform model tests on the original picture data set, the removed background data set, and the model trained on the mixed data set. It can be seen that the accuracy rate of the model

for the background removed data set and training on the mixed data set is 89.1%, and the recognition accuracy rate of the model trained with the background data set after the unified test is 3.91% lower than that of the model trained with the mixed data set, indicating that the use of the mixed data set to train the model has a better effect. After the mixed data set was enhanced, the training accuracy rate and the test accuracy rate were significantly improved, and the test accuracy rate reached 97.18%, indicating that the use of the mixed data set plus the data enhancement strategy has a better effect on model training.

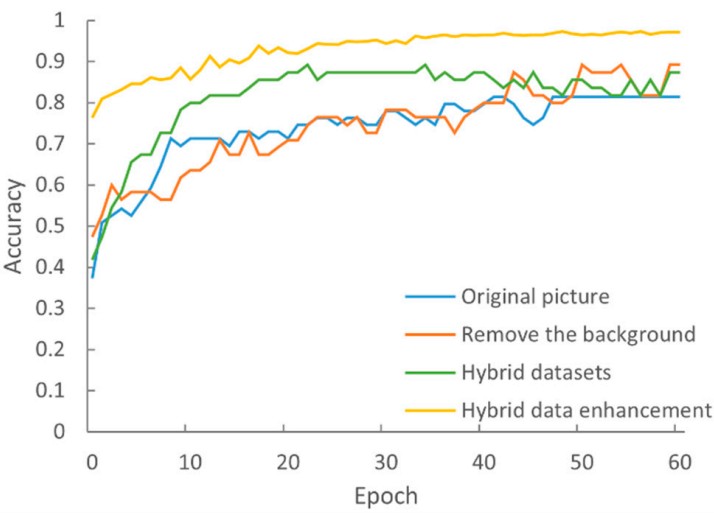

**Figure 7.** Comparison of the recognition accuracy of models for different data sets.

**Table 4.** The results of training models with different data sets.

| Data Set | Accuracy on the Training Data Set (%) | Accuracy on the Test Set (%) |
|---|---|---|
| Original image | 81.4 | 81.1 |
| Removed background | 89.1 | 86.8 |
| Mixed data | 89.1 | 90.7 |
| Mixed data enhanced | 97.3 | 97.2 |

*4.4. Comparative Test of Loss Function*

Figure 8 shows the test results obtained by training the model using different loss functions. It can be seen that the convergence effect when using PolyCrossEntropyLoss to train the model is the worst, the loss value is higher during the training process, and the convergence speed is slower. The loss value of the training model using the cross-entropy loss function is reduced, but the convergence speed is slower. The loss value of the FocalLoss training model is further reduced, the convergence speed is faster, and the training effect is better. Using PolyFocalLoss to train the model has the lowest loss value, faster convergence, and the best training effect.

Table 5 shows the results of training the original MobileNetV3-large network under different loss functions. It can be seen that the cross-entropy loss function of the original model is used to train the model. The recognition accuracy rate of the model is 97.18%, the accuracy rate is 97.17%, the recall rate is 97.2%, and the F1 score is 97.18. After using PolyCrossEntropyLoss to train the model, the model recognition accuracy rate increases by 0.63%, the accuracy rate and recall rate are both 97.83%, and the F1 score is 97.83. Using the focus loss function (FocalLoss) to train the model, the recognition performance of the model is further improved, the test accuracy rate reaches 98.18%, and the accuracy rate, recall rate, and F1 score are all improved. After PolyFocalLoss is used to train the model, the recognition accuracy rate of the model is the highest, reaching 98.45%, the accuracy rate is 98.48%, the recall rate is 98.45%, and the F1 score is 98.46.

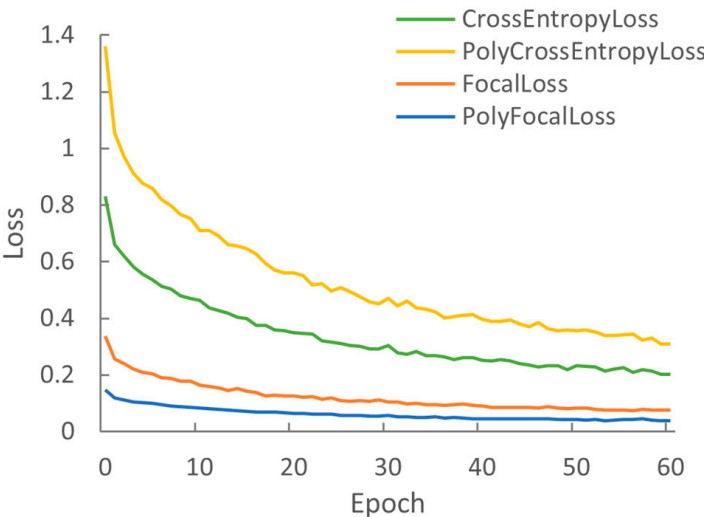

**Figure 8.** Comparison of training loss under different loss functions.

**Table 5.** The results of the training model under different loss functions.

| Loss Function | Accuracy (%) | Precision (%) | Recall (%) | F1 |
|---|---|---|---|---|
| CrossEntropyLoss | 97.18 | 97.17 | 97.20 | 97.18 |
| PolyCrossEntropyLoss | 97.81 | 97.83 | 97.83 | 97.83 |
| FocalLoss | 98.18 | 98.20 | 98.18 | 98.19 |
| PolyFocalLoss | 98.45 | 98.48 | 98.45 | 98.46 |

It can be seen that the loss value of the model during training under PolyCrossEntropyLoss is higher, but the recognition accuracy rate of the model after training under PolyCrossEntropyLoss is higher than that of the model trained under CrossEntropyLoss, indicating that PolyCrossEntropyLoss is more suitable for the distribution of the data set constructed in this experiment, which can better return the error and correct the weight in the process of back propagation. Using PolyFocalLoss to train the model has the best effect, increasing the recognition accuracy of the model by 1.27%, the accuracy rate and recall rate by 1.31% and 1.35%, respectively, and the F1 score by 1.28. It can be shown that improving the loss function is beneficial for improving the recognition performance of the model.

*4.5. Comparative Test of the Attention Mechanism*

Figure 9 shows the comparison test of the SE module and migration learning. It can be seen that the improved SE module improves the accuracy of the model training process more obviously. At the same time, when the model is trained without migration learning, the accuracy curve and loss curve of the training process oscillate obviously and converge slowly. When the model is trained with migration learning, the accuracy curve and loss curve during training are smooth and stable, and the convergence speed is faster.

In this paper, the attention mechanisms of CBAM [22], CA [23], and scSE [24] are also used to replace the original SE module, and the recognition performance of the model is compared through experiments. Table 6 lists the test results of different attention modules and migration learning strategies on the performance of model recognition. As can be seen from the ①② experiment, the use of the migration learning strategy helps to improve the recognition accuracy of the model, and the model recognition accuracy rate increases by 1.27%. As can be seen from the ②③④⑤⑥ test, the improved SE module has the most obvious effect on the performance of the model. The recognition accuracy of the model on the test set reaches 99.91%, which is 1.46% higher than before the improvement, and the model size is reduced by 4.3 M. By improving the attention SE mechanism in the original MobileNetV3-large network and introducing a migration learning strategy, the recognition accuracy rate of the model is 2.73% higher than before the improvement, while the model

size is reduced by 26.54%, indicating that the improvement method of this experiment is optimal.

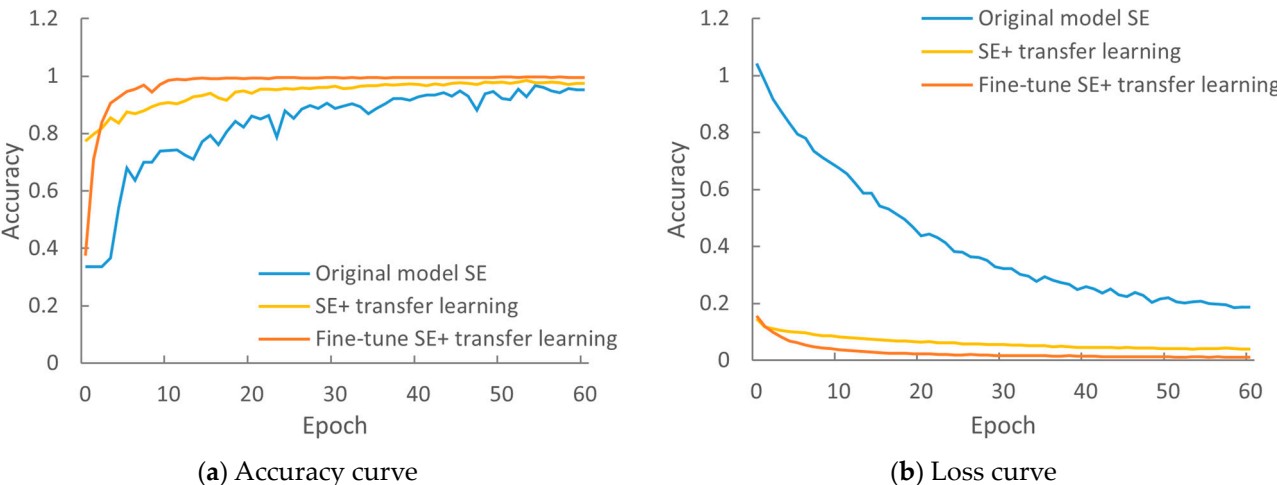

(**a**) Accuracy curve    (**b**) Loss curve

**Figure 9.** Comparative test results of the SE module and migration learning.

**Table 6.** Test results of the effects of different attention modules and migration learning strategies on model recognition performance.

| Test Number | Attention Module | Whether to Migrate to Learn | Accuracy (%) | Precision (%) | Recall (%) | F1 | Model Size (M) |
|---|---|---|---|---|---|---|---|
| 1 | SE | no | 97.18 | 97.19 | 97.20 | 97.19 | 16.2 |
| 2 | SE | yes | 98.45 | 98.48 | 98.45 | 98.46 | 16.2 |
| 3 | CBAM | yes | 99.64 | 99.64 | 99.64 | 99.64 | 11.9 |
| 4 | CA | yes | 99.09 | 99.10 | 99.09 | 99.09 | 11.2 |
| 5 | scSE | yes | 99.73 | 99.73 | 99.73 | 99.73 | 22.3 |
| 6 | Improved SE | yes | 99.91 | 99.91 | 99.91 | 99.91 | 11.9 |

### 4.6. Improved Model Evaluation

Figure 10 shows the confusion matrix of the original MobileNetV3-large network and the improved MobileNetV3-large network for the quality classification of shiitake mushrooms. The number of samples misclassified by the original network is 31, and the number of samples misclassified by the improved network is 1. Due to the different forms of third-class shiitake mushrooms, which include some second-class damaged shiitake mushrooms, when the damage is not obvious, it is easy to be confused with second-class shiitake mushrooms, so a prediction error occurs. The comparison shows that the improved MobileNetV3-large network classification effect is significantly improved.

As can be seen from Table 7, the model has a good effect on the recognition of various grades of shiitake mushrooms. The recognition accuracy rate, accuracy rate, and recall rate are all higher than 99.72%, and the F1 score is higher than 99.86. Among them, the recognition accuracy rate, accuracy rate, and recall rate of first-class shiitake mushrooms are all 100%, and the F1 score is 1. It can be shown that the improved model has high accuracy and good stability for the grade recognition of shiitake mushrooms in this test and that it can provide a reference for agricultural identification of shiitake mushroom grades.

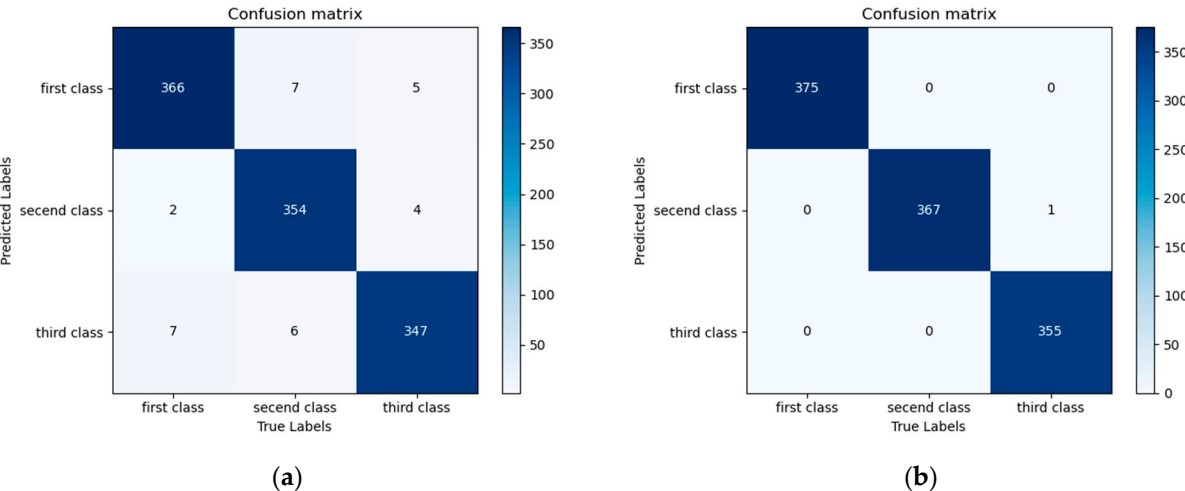

(**a**)                                                                 (**b**)

**Figure 10.** Original MobileNetV3-large (**a**) and improved MobileNetV3-large (**b**) confusion matrix diagram.

**Table 7.** Other evaluation indicators for improving the MobileNetV3-large network model.

| Grade Category | Accuracy (%) | Precision (%) | Recall (%) | F1 |
|:---:|:---:|:---:|:---:|:---:|
| First class | 100.00 | 100.00 | 100.00 | 100.00 |
| Second class | 100.00 | 99.72 | 100.00 | 99.86 |
| Third class | 99.72 | 100.00 | 99.72 | 99.86 |
| Avg | 99.91 | 99.91 | 99.91 | 99.91 |

*4.7. Comparison of Recognition Performance of Different Models*

In order to further verify the recognition performance of the improved model, this experiment compares the recognition results of different network models. As can be seen in Table 8, the recognition accuracy rate of the VGG16 network [25] is only 69.79%, the model size is 512 M, and the overall performance is poor. GoogLeNet [26], ResNet50 [27], and MobileNetV1 [28] exhibit slightly improved recognition accuracies, but the model size is larger than the improved network, and the overall performance is not high. Both the ShuffleNetV2×1 network [29] and the MobileNetV2 network are smaller in scale than the improved MobileNetV3 network, but the accuracy rate, accuracy rate, recall rate, and F1 score are significantly reduced. The recognition accuracy of EfficientNetV2-s [30] is higher than that of the original MobileNetV3-Large and slightly lower than that of the improved model, but the network scale is not dominant. In summary, the improved MobileNetV3 network recognition in this paper has high accuracy and a small network scale, which is suitable for mushroom quality classification tasks.

**Table 8.** Comparison results of recognition performance of different models.

| Model | Migrate to Learn | Accuracy (%) | Precision (%) | Recall (%) | F1 | Model Size (M) |
|:---:|:---:|:---:|:---:|:---:|:---:|:---:|
| VGG16 | yes | 69.76 | 73.84 | 69.92 | 71.83 | 512 |
| GoogLeNet | no | 74.04 | 82.04 | 74.35 | 78.01 | 38 |
| ResNet50 | yes | 82.24 | 72.90 | 82.24 | 77.29 | 90 |
| MobileNetV1 | no | 90.80 | 91.09 | 90.82 | 90.95 | 12.3 |
| MobileNetV2 | yes | 81.88 | 82.08 | 81.89 | 81.98 | 8.73 |
| MobileNetV3-Large | no | 97.18 | 97.19 | 97.20 | 97.19 | 16.2 |
| ShuffleNetV2×1 | yes | 71.58 | 71.83 | 71.58 | 71.70 | 4.95 |
| EfficientNetV2-s | yes | 97.45 | 97.48 | 97.45 | 97.46 | 77.8 |
| Improved MobileNetV3-Large | yes | 99.91 | 99.91 | 99.91 | 99.91 | 11.9 |

### 4.8. Visual Result Verification

The method shown in Figure 11 is implemented by gradient-weighted class activation mapping (Grad-CAM) [31]. After the improved MobileNetV3-large network model is recognized, the characteristic extraction and comparison results of each grade of shiitake mushrooms are obtained. It can be seen that the characteristics of the cap part of first-class shiitake mushrooms have received high attention from the model; the junction of the cap and the stipe of second-class shiitake mushrooms are the area of interest to the model; and the defective part of third-class shiitake mushrooms is the key location for the model to identify. The visualization results of feature extraction are consistent with the quality classification standards for shiitake mushrooms.

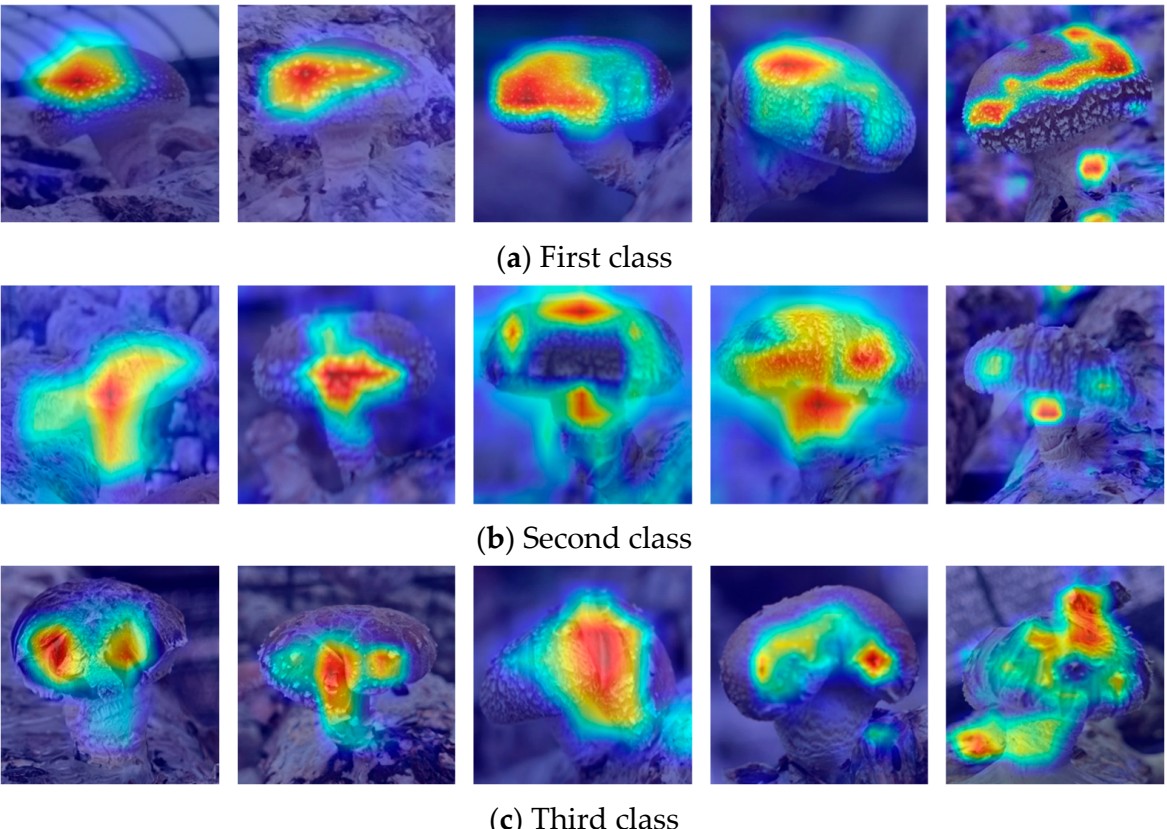

(**a**) First class

(**b**) Second class

(**c**) Third class

**Figure 11.** Comparison results of characteristic extraction of various grades of shiitake mushrooms.

### 4.9. Improvement of Mobilenet_V3 Compared to State-of-the-Art Methods

Table 9 demonstrates the results of comparing the recognition model in this paper with the current state-of-the-art recognition models. It can be seen that for different data sets, the authors have designed different recognition models, respectively. The data sets include Winter Jujube, Wheat, Mushrooom, Sauerkraut, and Pear. The recognition models are iResnet-50, D-VGG, and the recognition model with CNN architecture designed by the authors. Each designed model achieved better recognition results. After comparison, it can be seen that the improved MobilnetV3 designed in this paper performs well for mushroom quality classification. In addition, the recognition model designed in this paper is more robust because the collected data are captured in a real complex environment.

**Table 9.** Improvement of Mobilenet_V3 compared to state-of-the-art methods.

| Paper | Data Set | Classes | Images | Method/Model | Accuracy (%) | Precision (%) | Recall (%) |
|---|---|---|---|---|---|---|---|
| [32] | Winter Jujube | 5 | 20,000 | iResnet-50 | 98.35 | 98.40 | 98.35 |
| [33] | Wheat | 4 | 108 | ER-Stacking | 88.10 | 88.05 | 89.31 |
| [34] | Mushrooom | 6 | 6775 | D-VGG | 96.21 | 96.18 | 96.33 |
| [35] | Sauerkraut | 3 | 2190 | CNN | 95.3 | 93.2 | 92.9 |
| [36] | Pear | 3 | 398 | BP | 91.0 | 91.0 | 91.1 |
| Improved MobilnetV3 | Mushrooom | 3 | 10,991 | MobilnetV3 | 99.91 | 99.91 | 99.91 |

## 5. Conclusions

Deep learning techniques are widely used in precision agriculture. These techniques are used in various fields such as fruit and vegetable classification, face recognition, quality checking, yield estimation, disease prediction, etc. The success of these techniques has promoted the development of various types of deep learning models. This study is based on the MobilenetV3_large network architecture, which is improved and optimized. The application of recognizing and classifying the quality grades of shiitake mushrooms in the edible mushroom category is realized. By constructing a mixed data set, the model's performance of recognizing mushroom subjects in a complex background is improved. Based on the original MobilenetV3_large network, the recognition accuracy of the model is improved, and the network size is reduced by smaller changes. By improving the training strategy, the recognition accuracy of the network is improved, and the training cost is reduced. The experimental results show that the improved MobileNetV3_large network in this paper has better overall performance, with a recognition accuracy of 99.91% and a model size of 11.9 M. It can be used as a reference for the development of recognition and classification technology for shiitake mushrooms cultivated in greenhouse environments.

In future work, the deployment and application of the recognition model in end devices, e.g., cell phones, computers, intelligent robots, etc., will be further enhanced. At the same time, more varieties of mushrooms will be used to achieve a wider range of mushroom quality recognition and classification. In addition, this experimental model can be employed in different target detection networks to replace the classification network among them and compare the recognition results to help realize a wider range of related application research.

**Author Contributions:** F.Z., Y.S., Y.Z. and J.Q. conceived and designed the experiments; F.Z., Y.S. and J.Q. performed the experiments; F.Z., Y.S. and J.Q. analyzed the data and wrote the original manuscript; F.Z., Y.S., Y.Z., W.Z. and J.Q. reviewed and revised the manuscript. All authors have read and agreed to the published version of the manuscript.

**Funding:** This research was funded by the National Key Research and Development Program of China (2020YFD1000304-5), the Jilin Province Science and Technology Development Plan Project of China (20210202054NC) and the Jilin Province Science and Technology Development Plan Project of China (20220202029NC).

**Data Availability Statement:** The datasets generated and/or analyzed during the current study are available from the corresponding author upon reasonable request.

**Conflicts of Interest:** The authors declare no conflict of interest.

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
