# Peer review of "An Improved MobileNetV3 Mushroom Quality Classification Model Using Images with Complex Backgrounds"

_agronomy, doi:10.3390/agronomy13122924_

Round 1
Reviewer 1 Report
Comments and Suggestions for Authors
see the attachment

Reviewer 2 Report
Comments and Suggestions for Authors
The article needs a language revision, it is a good study and brings advances to the mushroom trade.
The introductory story goes well.
The materials and methods sections are clearly written.
The results are presented adequately, but the presentation of tables and images is confusing. For example, table 8, in my opinion, should be an figure.
The conclusions are motivated by the data obtained. Please indicate in the conclusion section how new the current study is and how this study differs from previously published articles on a similar topic.
References need to be standardized in terms of formatting.
Comments on the Quality of English Language
The article needs a language revision.
Reviewer 3 Report
Comments and Suggestions for Authors
Mushrooms are one of the components of the human diet. The population consumes both natural and human-grown mushrooms. Ensuring food security of the population requires improving mushroom production technologies, so the article is relevant. The article is devoted to the development of methods for sorting collected mushrooms by quality using computer technology. The article has a number of questions and suggestions for improving the presentation of research results.
1. At the beginning of the abstract of the article, it is necessary to briefly indicate its relevance. Now this is not obvious, but becomes clear only from the introduction.
2. Table 1 presents the criteria for classification of mushrooms. However, these are qualitative criteria that are determined visually by the collector and their use is to a certain extent subjective. From the point of view of the scientific component of the study, this is a weak side of the work.
3. Line 85 provides a link to table 1, although it should be to table 2.
4. It is not clear what the numbers in table 2 mean. Are the data sets the number of mushroom specimens?
5. Line 135 provides a link to table 2, although it should be to table 3.
6. What parameters are presented in Table 3? Units of measurement and explanations of the column names in the table are required.
7. Table 3 appears twice. It is necessary to revise the entire article regarding the numbering of tables, figures and references to them in the text. Errors are found throughout the article.
8. The list of references contains many references to conference abstracts that are not reviewed. References to articles in scientific journals are required.
9. The article describes the original method, but there are not enough results of its application in practice and an assessment of the economic effect of its implementation.
